# OpenReview forum: "Localized Concept Erasure in Text-to-Image Diffusion Models via High-Level Representation Misdirection"
_ICLR.cc/2026/Conference — ICLR 2026 Poster_

### Official Review · Reviewer_ARWM · 2025-10-27

**Soundness:** 2
**Presentation:** 2
**Contribution:** 2
**Rating:** 4
**Confidence:** 3

**Summary:**

This paper proposes HiRM, a concept erasure method that edits only the text encoder with a decoupled strategy: it sets the erasure target on the final text-encoder layer but updates only the first layer. There are two variants: HiRM-R (random target) and HiRM-S (semantic/safety target). Results on UnlearnCanvas and several adversarial benchmarks show a balanced trade-off: reasonable erasure, small utility loss, low compute, and transferable between model variants (e.g., SD/FLUX).

However, gains over Diff-Q on UnlearnCanvas are small, and there is a clear gap to SOTA on some adversarial benchmarks. The NSFW pipeline relies on templates and keyword gates, which may not generalize to other abstract concepts or compositional prompts.

Overall, the method is simple and practical, but the novelty is not enough and the robustness is not fully established.

**Strengths:**

- Decoupled supervision: HiRM applies the erasure objective at the final text-encoder block and only update the first block, avoiding low-level "representation shattering" and yielding a better erasure–utility trade-off.

- Efficiency and modularity: Only editing the text encoder makes the approach easy to train and transferable across pretrained T2I models (e.g., SD and FLUX variants) without modifying the backbone (UNets and Transformers).

- Balanced empirical profile: On UnlearnCanvas, HiRM-S matches or slightly improves over strong text-side baselines with very low cost rather than heavy retraining.

**Weaknesses:**

- The paper is an incremental iteration on Diff-Q: it still modifies the first transformer block of the text encoder but shifts the optimization target to the final layer, which carries higher-level representations. Conceptually sound, but not a paradigm shift.

- Gains over Diff-Q are small. In Table 2, IRA (object erasure) drops from 98.37 to 98.18 (−0.19 pts). The gap to SOTA remains large: in Table 3 the method underperforms on UnlearnDiffAtk (19.01 vs. 9.80) and MMA-Diffusion (3.30 vs. 0.40).

- For NSFW, HiRM-S constructs a safety vector by differencing prompt embeddings, making it dependent on template and vocabulary. This may risk brittleness under template changes, across languages, or at nuanced concept boundaries.

**Questions:**

1. Your method uses two pipelines for *nudity* vs. *non-nudity* (keyword/template gated). Why is this split necessary? Could a single pipeline (no keyword gate) achieve similar or better results?

2. For abstract attributes (e.g., *violent scene* vs. *scene*), the difference vector seems to capture a theme (e.g., war) rather than the attribute(violence). How do you ensure the vector encodes the attribute itself, not a topic proxy? Show stress tests across domains (e.g., domestic/school violence).

3. How does the method handle compositional prompts with scope and negation (e.g., "… but without weapons")? Can you provide some targeted evaluations (AND/OR/NOT, "topic vs. attribute" disentanglement).

---

> ### Author Response · Authors · 2025-11-23
> **Official Response to Reviewer ARWM (1/4)**
>
> > [W1] The paper is considered an incremental iteration on Diff-Q that shifts the optimization target to the final layer, which is conceptually sound but not a paradigm shift.
>
> We agree that our method is conceptually close to Diff-Q in that both operate on the first block of the text encoder, and we do not claim a paradigm shift. Our contribution is to refine this class of text-side erasure methods by showing that **where the erasure objective is applied and where parameters are updated matters significantly** for diffusion text encoders.
>
> Motivated by prior analyses of CLIP-style encoders (e.g., Toker et al., 2024), which show that high-level semantics are formed in the last layers while early layers encode lower-level causal states, HiRM applies the misdirection loss at the **final block** but restricts updates to the **early blocks**. This decoupled “final-block steering + early-block update” design is not present in Diff-Q (which effectively steers and updates at the first block) and is crucial for erasing high-level concepts without globally shattering the representation. Our ablations in Table 7 support this: applying the objective only to early blocks severely underperforms for style concepts, while targeting the final block yields much stronger erasure.
>
> In addition, this design leads to **qualitatively different behavior** on more challenging settings. On UnlearnCanvas (Table 2), Diff-Q and HiRM can look similar in aggregate metrics, but on broader and safety-critical benchmarks (COCO retention (Table 1), NSFW/adversarial evaluations, and zero-shot transfer from SD-1.4 to Flux (Table 4))Diff-Q exhibits substantially worse non-target preservation and robustness. HiRM is not a paradigm shift over Diff-Q, but a targeted refinement of text-side concept erasure that, through a decoupled loss/update design informed by prior analyses, achieves a meaningfully better erasure–retention–robustness trade-off on hard T2I safety benchmarks.

---

> ### Author Response · Authors · 2025-11-23
> **Official Response to Reviewer ARWM (2/4)**
>
> > [W2] Gains over Diff-Q are small, and the gap to SOTA remains large as the method underperforms on benchmarks like UnlearnDiffAtk and MMA-Diffusion.
>
> We appreciate the reviewer’s detailed analysis of the performance margins. We agree that, on UnlearnCanvas (Table 2), the improvement over Diff-Q in IRA/CRA is numerically small. Our intention is not to claim a large gain on this specific metric, but to show that HiRM provides a **better overall balance between safety and utility across different benchmarks**.
>
> For Diff-Q, the scores on UnlearnCanvas appear close to HiRM, but the difference becomes clear on more challenging evaluations. As shown in Table 3, Diff-Q is much more **vulnerable to adversarial attacks**(higher Ring/MMA/MU), whereas HiRM-S **reduces the attack success rates substantially**. In addition, Diff-Q causes a larger drop in COCO quality (COCO CLIP: 0.289), while HiRM-S maintains a higher score (0.306). This indicates that the gap between the two methods is not limited to a 0.19-point difference on IRA, but also **appears in adversarial robustness and preservation of general image quality**.
>
> Regarding the gap to methods such as SalUn and RECE on UnlearnDiffAtk and MMA-Diffusion, we see this as a consequence of a **safety–utility trade-off**. SalUn and RECE achieve very strong defense, but at **the cost of noticeably reducing the model’s generative quality**, as reflected in their COCO CLIP scores (SalUn: 0.293, RECE: 0.277 vs. 0.315 for the original model). By contrast, HiRM is designed to **keep the COCO CLIP score close to the original** (0.306 vs. 0.315), while still **providing strong protection against attacks**.
>
> Finally, we found that HiRM **complements U-Net–based defenses rather than directly competing with them**. When we combine a HiRM-tuned text encoder with denoiser-based methods (CA, ESD, EraseAnything), we observe **consistent improvements in safety metrics without harming COCO performance**. The table below summarizes these results:
>
>
>
> | Method | Ring-16 ↓ | Ring-38 ↓ | Ring-77 ↓ | MMA ↓ | I2P ↓ | COCO-1k (CLIP) ↑ |
> | :--- | :--- | :--- | :--- | :--- | :--- | :--- |
> | **CA** | 11.58 | 11.58 | 6.32 | 3.29 | 1.83 | **0.302** |
> | **CA + HiRM-R** | **3.16%** | **2.11** | **2.11%** | 4.40 | **1.11** | **0.302** |
> | | | | | | | |
> | **ESD** | 41.05 | 33.68 | 32.63 | 7.10% | 2.93 | **0.307** |
> | **ESD + HiRM-R** | **12.63** | **7.37** | **4.21** | **3.30** | **2.51** | 0.306 |
> | | | | | | | |
> | **EraseAnything** | 29.47 | 24.21 | 26.32 | 6.60 | 2.64 | **0.305** |
> | **EA + HiRM-R** | **3.16** | **1.05** | **3.16** | **2.50** | **1.57** | **0.305** |
>
> These results show that adding an encoder-side HiRM patch can substantially improve the safety metrics of existing U-Net–based methods, while keeping COCO-1k CLIP scores almost unchanged. We have included this discussion and table in Section 5.3 of the revised manuscript to emphasize that HiRM provides **a lightweight, transferable encoder component that improves robustness** both on its own and when combined with strong denoiser-based approaches.
>
> ---
>
> >[W3] For NSFW, HiRM-S constructs a safety vector by differencing prompt embeddings, making it dependent on template and vocabulary, which may risk brittleness under template changes, across languages, or at nuanced concept boundaries.
>
> We agree that constructing a safety direction by differencing prompt embeddings can, in principle, be sensitive to the choice of templates and vocabulary. To mitigate this, we follow the multi-template estimation strategy used in Ring-A-Bell: instead of relying on a single prompt pair, we **average embedding differences over a diverse set of templates**. This is a common practice in recent work and is intended to capture a more general semantic direction for the attribute rather than overfitting to a specific phrasing.
>
> Averaging over multiple templates **helps reduce brittleness**, because the resulting safety vector is influenced by many paraphrases and contexts rather than a single wording. In the revision, we have made this design choice explicit and provided a **stress-test analysis** (see our response to [Q2]), where we evaluate HiRM-S under varied templates and nuanced NSFW boundaries (e.g., milder vs. more explicit prompts). In these tests, HiRM-S **remains robust while preserving non-target content**.
>
> We also examined the dependency on template sets more directly. In an additional experiment on five style concepts, we varied the number of templates (10, 30, and 50) used to estimate the direction and measured unlearning accuracy (UA) and retention (IRA/CRA). As shown in the table below, **the performance differences across template counts are small and do not change the overall conclusions**:
> | Number of Templates | UA | IRA | CRA |
> | :--- | :---: | :---: | :---: |
> | **10** | 95.0 | 93.42 | 98.28 |
> | **30** | 100.0 | 93.58 | 98.06 |
> | **50** | 100.0 | 93.14 | 97.92 |

---

> ### Author Response · Authors · 2025-11-23
> **Official Response to Reviewer ARWM (3/4)**
>
> >[Q1] Your method uses two pipelines for nudity vs. non-nudity (keyword/template gated). Why is this split necessary? Could a single pipeline (no keyword gate) achieve similar or better results?
>
> We thank the reviewer for this constructive question. Originally, we employed a dual-pipeline approach because **defining explicit keywords for abstract concepts like nudity is inherently challenging**, whereas concrete concepts (e.g., specific styles) are **easily definable by keywords**.
>
> Following the reviewer’s suggestion, we **evaluated the feasibility of a unified pipeline** by applying the template-gated mechanism (originally designed for nudity) to style concepts. As demonstrated in the table below, we observed **no significant performance difference** compared to our original keyword-gated approach.
>
> | Concept | UA (HiRM-S) | IRA | CRA | UA (HiRM-Template-50) | IRA | CRA |
> | :--- | :---: | :---: | :---: | :---: | :---: | :---: |
> | **Bricks** | 100.0 | 94.6 | 98.3 | 100.0 | 89.4 | 97.9 |
> | **Cartoon** | 100.0 | 91.4 | 97.4 | 100.0 | 92.5 | 97.3 |
> | **Crayon** | 100.0 | 92.2 | 97.6 | 100.0 | 91.4 | 97.2 |
> | **Van_Gogh** | 95.0 | 94.2 | 97.7 | 100.0 | 96.0 | 98.4 |
> | **Dapple** | 100.0 | 94.3 | 99.0 | 100.0 | 96.4 | 98.8 |
>
> This empirically confirms that while a **single unified pipeline is feasible**, the choice of gating mechanism does not critically impact performance. Thus, if keywords are easily definable for a target concept, the **keyword-gated approach remains a valid and efficient option**. We appreciate this valuable suggestion, as it **demonstrates the flexibility of our framework**. We have incorporated these findings into Appendix B.7 of the revised manuscript.
>
> ---
>
> > [Q2] For abstract attributes (e.g., violent scene vs. scene), the difference vector seems to capture a theme (e.g., war) rather than the attribute(violence). How do you ensure the vector encodes the attribute itself, not a topic proxy? Show stress tests across domains (e.g., domestic/school violence).
>
> We agree with the reviewer that, for abstract attributes such as nudity or violence, a simple difference vector can partially entangle the attribute with specific topics (e.g., bedroom selfies or porn-like scenes). In our NSFW pipeline, we do not claim that the safety vector fully disentangles "nudity" from all possible contexts; rather, we adopt the Ring-A-Bell style multi-template estimation procedure as a pragmatic approximation, which yields an empirical direction vector intended to capture the target concept (e.g., “nudity”).
>
> Although the specific safety vector estimation is a **plug-in component rather than our core contribution** (which is the decoupled encoder editing framework), we recognize the importance of verifying whether the vector captures the attribute or merely a topic proxy.
>
> To address this, we conducted a **stress test across diverse domains**, evaluating erasure performance on four topic (Beach, Renaissance, Shower, Bathroom). Using GPT-5, we generated prompts containing the word “nude” for each topic (eg., "A person sunbathing nude on a beach").
>
> Qualitative results have included in Figure 21 of the revised manuscript. As illustrated, HiRM-S **effectively eliminates nudity across all four diverse topics** (Beach, Renaissance, Shower, Bathroom) while **successfully preserving the core thematic context**.
>
> In conclusion, our stress tests confirm that HiRM **selectively removes the nudity attribute while preserving the semantic context, even in entangled scenarios** (e.g., Beach, shower scenes).

---

> > ### Author Response · Authors · 2025-11-23
> > **Official Response to Reviewer ARWM (4/4)**
> >
> > >[Q3] How does the method handle compositional prompts with scope and negation (e.g., "… but without weapons")? Can you provide some targeted evaluations (AND/OR/NOT, "topic vs. attribute" disentanglement).
> >
> > We acknowledge that compositional prompts with explicit scope and negation present a challenge, as HiRM operates on representations from the underlying text encoder and thus inherits its logical reasoning characteristics without a separate module.
> >
> > To empirically assess this behavior, we conducted a **targeted evaluation using GPT-5 to generate distinct prompt variations**, 'Base', 'And' (Topic + Attribute), and 'Not' (Negation), across scenarios such as 'Beach' and 'Bedroom'. For instance, given a 'Base' prompt such as 'A couple walking hand in hand along the shoreline of a quiet beach', we generated an 'And' variation (e.g., 'A nude couple...' ) and a 'Not' variation (e.g., '...without nudity' ).
> >
> > As illustrated in Figure 22 in revised manuscript, HiRM **demonstrates effective attribute disentanglement** in these compositional settings. In the 'And' case (e.g., combining a beach scene with nudity), the method **successfully erases the visual nudity** (MLLM Safety Score: 5) while **preserving core background elements** like the sunset and beach, confirming that it **targets the specific attribute rather than suppressing the entire scene**. Furthermore, in the 'Not' case (e.g., "without nudity"), HiRM respects the negation by generating fully clothed individuals and maintaining a composition identical to the Base topic.
> >
> > Overall, these targeted tests suggest that HiRM can **distinguish, in many practical cases, between positive inclusion (“with nudity”) and negative contexts (“without nudity”)** without severe over-erasure of the underlying topic. We have added this qualitative examples to the appendix of the revised manuscript.

---

> ### Comment · Reviewer_ARWM · 2025-11-27
> **Reply to rebuttal**
>
> Thank you for the detailed rebuttal that has addressed my concerns. I have raised my score accordingly.

---

> > ### Author Response · Authors · 2025-11-28
> > **Official Comment by Authors**
> >
> > Dear Reviewer ARWM,
> >
> > We thank the reviewer for the positive feedback. We are glad that we have addressed your concerns. We would be grateful for any further questions or comments you might have.
> >
> > Best regards,
> >
> > The Authors

---

### Official Review · Reviewer_955x · 2025-10-30

**Soundness:** 3
**Presentation:** 3
**Contribution:** 3
**Rating:** 6
**Confidence:** 3

**Summary:**

The paper proposed a concept erasure method that operates solely on the text encoder, called High-Level Representation Misdirection (HiRM). Specifically, HiRM guides the token representations output by the last layer of the text encoder in an incorrect direction and updates the weights of the first layer accordingly. Extensive experiments demonstrate the effectiveness of the proposed method.

**Strengths:**

* The writing is fluent and logically coherent, exhibiting strong readability.
* The proposed method is highly modular, requiring only a modification to the first layer of the text encoder to achieve concept erasure, which demonstrates strong practical applicability.
* The experimental design is thorough, and the results yield insights with meaningful implications for the research community.

**Weaknesses:**

* The proposed method is relatively empirical and experimental, lacking solid theoretical support. It would be more beneficial to the community if the interpretability of the erased concept could be analyzed from the perspective of the distribution of activated neurons.
* The core idea of HiRM lies in computing the loss based on the output of the last layer of the text encoder, thereby enhancing its ability to erase high-level concepts. However, in Figure 2, the elimination of the high-level concept *Fauvism* appears to be less effective than that of the more concrete concept *Tree*. Why is this the case?
* Although the authors provide additional examples in the Appendix, the reviewer finds that the visualizations still lack sufficient diversity. Including more prompts and results across a wider range of backbone models would make the work more convincing.
* The reviewer observes that both HiRM-R and HiRM-S may, to some extent, affect the similarity between non-target concepts and the original model outputs — as shown in Figures 9 and 10. Could the authors please provide an explanation for this issue?

**Questions:**

See 'Weaknesses'.

---

> ### Author Response · Authors · 2025-11-23
> **Official Response to Reviewer 955x (1/1)**
>
> >[W1] The proposed method is relatively empirical and lacking solid theoretical support, so it would be beneficial to analyze the interpretability of the erased concept from the perspective of the distribution of activated neurons.
>
> We appreciate the reviewer’s constructive suggestion to enhance the interpretability of our method by analyzing the distribution of activated neurons. We measured the Jaccard Similarity (quantifying the overlap ratio of the most active neuron indices between the original and unlearned models) of the Top-50 activated neurons; results show that the target concept ('Van Gogh') **undergoes a drastic shift in activation patterns** (Similarity: 0.0870 in First Block, 0.1364 in Final Block), whereas **non-target concepts remain stable** (Average: 0.4818 and 0.5525, respectively). This pattern suggests that HiRM **primarily disrupts the representation subspace associated with the erased concept** at the intervention site, while leaving the activation distribution for unrelated concepts comparatively intact. We have included this analysis and the corresponding figure in Section 5.1 and Appendix B.3 of the revised manuscript.
>
> ---
>
> >[W2] Although HiRM aims to enhance the ability to erase high-level concepts, the elimination of the high-level concept Fauvism in Figure 2 appears to be less effective than that of the more concrete concept Tree.
>
> We thank the reviewer for this insightful observation regarding the t-SNE visualization. We agree that, at first glance, the shift in the “Fauvism” plot may appear less pronounced than in the “Tree” plot. However, this is largely a **limitation of the visualization**: for each concept and model we run t-SNE separately, and t-SNE is designed to preserve local neighborhoods rather than absolute geometry. As a result, the apparent displacement and scale in different 2D projections are **not directly comparable across plots**, and **visual distances do not faithfully reflect the relative magnitude of erasure**.
>
> To avoid over-interpreting t-SNE, we rely on quantitative metrics. In Table 2, HiRM-R actually achieves higher unlearning accuracy (UA) for style concepts than for object concepts. For the specific examples in Figure 2, HiRM-R attains 100% UA for “Fauvism” and 80% UA for “Tree”, indicating that the **style concept is in fact erased more completely** despite the less dramatic visual shift in the 2D embedding.
>
> ---
>
> > [W3] The reviewer finds that the visualizations still lack sufficient diversity, so including more prompts and results across a wider range of backbone models would make the work more convincing.
>
> We fully agree with the reviewer that more diverse visualizations would strengthen the paper. To address this, we commit to adding comprehensive qualitative results covering diverse backbones (e.g., Flux) and prompts. We have included these examples in Appendix B.2 of the revised manuscript.
>
> ---
>
> > [W4] The reviewer observes that both HiRM-R and HiRM-S may affect the similarity between non-target concepts and the original model outputs as shown in Figures 9 and 10, requesting an explanation for this issue.
>
> We appreciate the reviewer’s observation. Fundamentally, the objective of concept erasing is to eliminate the target concept while minimizing the influence on non-target concepts. The phenomenon highlighted by the reviewer represents an **intrinsic trade-off** in achieving this goal. However, despite these visual shifts, we emphasize that the **semantic integrity of non-target concepts is preserved**. This is more importantly, the quantitative results in Table 2, where HiRM achieves over 95% Retention Accuracy (IRA/CRA). This shows that classifiers successfully recognize non-target concepts despite minor visual changes, confirming that the core **semantics remain intact**.

---

> > ### Comment · Reviewer_955x · 2025-11-27
> >
> > Thank you. The authors’ response has addressed the concerns I raised, and I will maintain my positive score.

---

> > > ### Author Response · Authors · 2025-11-28
> > > **Official Comment by Authors**
> > >
> > > Dear Reviewer 955x,
> > >
> > > Thank you for reviewing our responses. We sincerely appreciate the opportunity to address your concerns and improve our study based on your feedback. We are pleased that our clarifications resolved most of your concerns.
> > >
> > > If you have any remaining questions or suggestions, we would greatly appreciate your comments to further strengthen our study
> > >
> > > Best regards,
> > >
> > > The Authors

---

### Official Review · Reviewer_Jypt · 2025-10-30

**Soundness:** 1
**Presentation:** 3
**Contribution:** 2
**Rating:** 4
**Confidence:** 3

**Summary:**

This paper leverages a fine-tuned text encoder for concept erasure. The authors propose fine-tuning the early layers of the text encoder by directing the high-level semantic representations of target concepts toward random or super-category directions. Experimental results on single-concept erasure demonstrate that the method achieves comparable performance while requiring minimal time and memory.

**Strengths:**

1. Rather than training the diffusion model parameters, the authors fine-tune the text encoder parameters to improve efficiency.
2. The proposed method is straightforward and easy to implement.
3. The idea of using high-level semantic representations to guide updates in the early layers is interesting.

**Weaknesses:**

1. Related work is missing. SPEED [A] leverages null-space constraints to achieve rapid concept erasure and can be extended to multi-concept scenarios. The authors should include SPEED as a baseline and compare efficiency.

2. Although the authors mention plans to extend the proposed method to multi-concept erasure, its tuning-based nature limits scalability. As more concepts are introduced, optimizing the early layers becomes increasingly difficult. Therefore, the authors should include a multi-concept erasure setting to demonstrate the method’s potential.

3. Table 2 shows that Diff-Q achieves comparable results on UnlearnCanvas, and unlike training-based methods, it attains higher performance via a closed-form solution. It shows limited improvement introduced by HiRM. Besides, Table 1 reports the low performance of Diff-Q in preserving untargeted concepts. Does this suggest that the metrics used in Table 2 may not effectively reflect the ability to preserve untargeted concepts, given that IRA and CRA scores are near 100?

[A] Li, Ouxiang, et al. "Speed: Scalable, precise, and efficient concept erasure for diffusion models." arXiv preprint arXiv:2503.07392 (2025).

**Questions:**

1. Comparison with SPEED.
2. Support for multi-concept erasure.
3. Choice and justification of evaluation metrics.
4. Would introducing additional constraints on the higher layers (e.g., the last two layers) improve performance?

---

> ### Author Response · Authors · 2025-11-23
> **Official Response to Reviewer Jypt (1/3)**
>
> > [W1] Related work is missing; the authors should include SPEED as a baseline and compare efficiency as it leverages null-space constraints to achieve rapid concept erasure.
>
> > [W2] Since the tuning-based nature limits scalability as optimizing early layers becomes increasingly difficult, the authors should include a multi-concept erasure setting to demonstrate the method’s potential.
>
> ## **Response of W1 and W2**
>
> In summary, we now (i) include SPEED as a baseline and compare both **efficiency and adversarial robustness**, and (ii) provide preliminary evidence that HiRM can be extended to **multi-concept settings**, as well as a **hybrid SPEED+HiRM configuration** that combines the strengths of both methods.
>
> We thank the reviewer for highlighting SPEED [A] and for raising the scalability question. Our paper primarily focuses on single-concept erasure, which we consider indispensable for safety-critical applications such as nudity removal, where highly precise and robust elimination of a specific concept is required. That said, we agree that multi-concept erasure is an important challenge, and we address both points jointly below.
>
> ### **Multi-concept erasure with HiRM**
>
> To demonstrate HiRM’s potential for multi-concept erasure ([W2]), we first conducted an experiment on UnlearnCanvas by learning individual LoRA modules for 6 different style concepts and merging them via simple weight averaging.
>
> These results suggest that our decoupled strategy can be extended to multi-concept scenarios: **average unlearning accuracy reaches 88.33% with high cross-domain accuracy (98.8%)**. At the same time, in-domain accuracy drops to 65.56% due to the simplicity of the fusion procedure (plain averaging). We view this as an expected limitation of the naive combination rather than of HiRM itself, and anticipate that more sophisticated module-fusion techniques (e.g., learned weighting, conflict-aware fusion) could mitigate this trade-off. We have included this experiment in Appendix B.6 of the revised manuscript.
>
> | Metric            | Accuracy (%) |
> |-----------------------------|--------------|
> | Average Unlearning Accuracy | 88.33        |
> | In-domain Accuracy          | 65.56        |
> | Cross-domain Accuracy       | 98.80        |
>
> ### **Comparison with SPEED**
>
> Regarding the comparison with SPEED ([W1]), we evaluated HiRM-S against SPEED using the official SPEED weights provided by the authors. While SPEED is explicitly designed to **prioritize scalability and performance for mass erasure** (e.g., ~5s for 100 concepts owing to its closed-form formulation), HiRM-S is conversely **tailored for efficient and robust single-concept erasure** (1.2s vs. 3.6s for SPEED). The performance reported for SPEED in Table 2 of [A] also indicates that it is substantially more effective than HiRM-S when erasing a large number of concepts simultaneously, which is consistent with SPEED’s design goal.
>
> However, we also observe that SPEED and HiRM-S exhibit **different behavior under adversarial attacks**, particularly for safety-sensitive nudity removal. The following table compares the two methods on nudity-targeted adversarial benchmarks and COCO:
>
> | Method | Ring-16 ↓ | Ring-38 ↓ | Ring-77 ↓ | MMA ↓ | MU ↓ | I2P ↓ | COCO-10k (CLIP) ↑ | COCO-10k (FID) ↓ | Speed (s)|
> | :--- | :---: | :---: | :---: | :---: | :---: | :---: | :---: | :---: | :---: |
> | **SPEED [A]** | 43.16 | 42.11 | 49.47 | 20.80 | 59.15 | 1.55 | **0.309** | 6.78 | 3.6 |
> | **HiRM-S (Ours)** | **1.05** | **1.05** | **0.00** | **3.30** | **19.01** | **0.66** | 0.306 | **6.75** | **1.2** |
>
> HiRM-S achieves substantially better resistance to nudity-targeted adversarial prompts, while maintaining comparable COCO-10k CLIP/FID scores and lower runtime. This reflects a design difference: **SPEED emphasizes large-scale, multi-concept scalability**, whereas **HiRM-S emphasizes robust single-concept erasure with strong adversarial robustness** and minimal encoder updates.

---

> ### Author Response · Authors · 2025-11-23
> **Official Response to Reviewer Jypt (2/3)**
>
> ### **Hybrid SPEED + HiRM for multi-concept & robustness**
>
> Building on the synergy observed with other U-Net–based methods in our response to Reviewer X6qh, we further examined whether **HiRM and SPEED are complementary** in multi-concept scenarios. To this end, we combined:
> * a SPEED-optimized U-Net (erasing 50 celebrity identities), and
> * a HiRM-S-optimized text encoder (erasing nudity),
>
> and evaluated whether this hybrid configuration preserves SPEED’s multi-concept erasure while inheriting HiRM’s adversarial robustness. The results are as follows:
>
>
> | Method          | Ring-16 ↓ | Ring-38 ↓ | Ring-77 ↓ | MMA ↓  | I2P ↓    | COCO-10k (CLIP) ↑ | COCO-10k (FID) ↓ | ACC$_e$ | ACC$_r$ |
> | :-------------- | :-------: | :-------: | :-------: | :----: | :------: | :---------------: | :--------------: | :---: | :---: |
> | **SPEED + HiRM** | 1.05 | 1.05 | 2.11   | 1.70 | 0.43 | 0.304             | 7.43             | 3.64  | 79.30 |
>
>
> Here, ACC$_e$ denotes an erasure error metric for the 50-celebrity task (lower is better), and ACC$_r$ denotes retention accuracy (higher is better). The hybrid model:
> * achieves very low Ring-16/38 and MMA = 1.70, indicating **strong robustness** in the nudity dimension, and
> * **maintains a low erasure error** (ACC$_e$ = 3.64%) for the 50 celebrities, with retention accuracy (ACC$_r$) of 79.30%.
>
> We do observe a modest compromise in COCO utility (COCO-10k CLIP decreases slightly from 0.309 for SPEED alone to 0.304, with a mild FID increase), and a drop in retention accuracy for the 50 celebrities (from 88.48% to 79.30%). Nonetheless, the overall behavior shows that combining SPEED (U-Net) and HiRM (text encoder) is feasible and **yields a hybrid solution that inherits SPEED’s multi-concept capability while substantially improving adversarial robustness**.
>
> We have incorporated SPEED into the related work (Section 2) and experiments (Section 5.3), acknowledging its scalability for multi-concept erasure. Simultaneously, we emphasize that HiRM provides a **complementary encoder-side mechanism** that is particularly effective for robust single-concept safety (e.g., nudity) and for hybrid scenarios where encoder and U-Net edits are combined.
>
> ---
>
> >[W3] Table 2 shows that Diff-Q achieves comparable results, suggesting limited improvement by HiRM, and given its low preservation in Table 1, it is questioned whether the metrics in Table 2 effectively reflect the ability to preserve untargeted concepts.
>
> We appreciate the reviewer’s careful observation. The apparent discrepancy indeed comes from the fact that Tables 1 and 2 probe different domains with different notions of “retention.”
>
> In Table 2 (UnlearnCanvas), IRA and CRA are **classifier-based metrics** trained specifically on a narrow set of prompts and concepts in that benchmark. They effectively measure how often the erased model and the original model agree on non-target classifications within this controlled distribution. In contrast, Table 1 evaluates retention on **COCO-style, open-domain prompts** using CLIP similarity between generated images and their captions, which captures broader perceptual alignment rather than classifier agreement on a fixed label set.
>
> As a result, a method like Diff-Q can look almost perfectly aligned on IRA/CRA (near-100% agreement on UnlearnCanvas classifiers) while still showing noticeable degradation on COCO/CLIP and other settings where prompts are more complex and diverse. This is not unique to Diff-Q: we observe similar behavior for other methods such as SalUn and RECE, which perform well on UnlearnCanvas metrics but exhibit weaker retention or robustness under COCO and adversarial evaluations.
>
> We therefore do not interpret IRA/CRA as a complete measure of non-target preservation, but rather as one piece in a broader evaluation. To obtain a more faithful picture, we complement UnlearnCanvas metrics with COCO FID/CLIP, NSFW/adversarial attack success rates, and MLLM-as-a-judge scores on selected datasets, which align better with human perception. Across this richer set of metrics, HiRM provides a consistently better erasure–retention–robustness trade-off than Diff-Q, even if the improvement on IRA/CRA alone appears modest.

---

> ### Author Response · Authors · 2025-11-23
> **Official Response to Reviewer Jypt (3/3)**
>
> > [Q1] Comparison with SPEED.
>
> Please refer to our response to [W1], where we provide a detailed comparison with SPEED regarding robustness and efficiency.
>
> ---
>
> > [Q2] Support for multi-concept erasure.
>
> Please refer to our response to [W2], where we explicitly demonstrate HiRM’s scalability to multi-concept scenarios using the UnlearnCanvas benchmark and LoRA merging techniques.
>
> ---
>
> > [Q3] Choice and justification of evaluation metrics.
>
> Please refer to our response to [W3], where we address the potential discrepancy between tables and justify our holistic evaluation approach (combining classifiers, perceptual metrics, and MLLM-as-a-judge) to ensure a balanced assessment of erasure and utility.
>
> ---
>
> > [Q4] Would introducing additional constraints on the higher layers (e.g., the last two layers) improve performance?
>
> To address this question, we conducted an additional experiment (HiRM-S-const) by introducing constraints to the last two layers. Specifically, HiRM-S-const extracts nudity-related representations from each of the final two layers and misdirects them using the safety vector derived by differencing prompt embeddings. As shown in the table below, while this modification yielded marginal improvements in utility preservation (improving FID from 6.75 to 6.00), it resulted in a significant degradation in unlearning performance.
>
> Specifically, the MMA score surged from 3.30% to 24.10%, and the Ring-16 vulnerability escalated from 1.05% to 5.26%. Given the critical importance of robustness against adversarial attacks, the marginal gain in utility is insignificant.
>
> | Model           | Ring-16 ↓ | Ring-38 ↓ | Ring-77 ↓ | MMA ↓ | I2P ↓ | COCO-10k (CLIP) ↑ | COCO-10k (FID) ↓ |
> |-----------------|---------|---------|---------|--------|--------|----------------|---------------|
> | **HiRM-S-const** | 5.26   | **0.00**   | 1.05   | 24.10 | 1.02 | **0.308** | **6.00**|
> | **HiRM-S** | **1.05** | 1.05 | **0.00** | **3.30** | **0.66** | 0.306 | 6.75 |
>
> ---
>
> [A] Li, Ouxiang, et al. "Speed: Scalable, precise, and efficient concept erasure for diffusion models." arXiv preprint arXiv:2503.07392 (2025).

---

> ### Author Response · Authors · 2025-11-28
> **Official Response to Reviewer Jypt**
>
> Dear Reviewer Jypt,
>
> Thank you once again for your insightful and constructive suggestions on our submission. As the discussion period draws to a close, we would be very grateful to know if you have any remaining questions or final thoughts. Thank you for your time and valuable contributions to improving our work.
>
> Best regards,
>
> The Authors

---

### Official Review · Reviewer_X6qh · 2025-11-02

**Soundness:** 2
**Presentation:** 2
**Contribution:** 2
**Rating:** 4
**Confidence:** 4

**Summary:**

The paper is motivated by the observation that Diff-QuickFix, which modifies only the first self-attention layer, performs poorly when addressing high-level abstract concepts such as nudity or NSFW content. Moreover, when modifying the entire first transformer block toward random vectors (inspired by the RMU approach in LLMs), it tends to over-unlearn, negatively affecting unrelated concepts.

To address this, the paper proposes targeting high-level concept representations in the final encoder block (rather than the first layer) while applying updates only to the early-layer weights. This decoupling strategy enables more precise concept removal with minimal impact on unrelated concepts.

**Strengths:**

•  The paper is clearly written and easy to follow.

•  The proposed method is simple and intuitively reasonable.

•  The experiments appear comprehensive, and the results look promising.

**Weaknesses:**

-	In diffusion model training, the text encoder is typically a pre-trained model (e.g., CLIP text encoder) and remains frozen throughout the training process. This means that if unlearning is applied only to the text encoder, malicious users could easily replace the sanitized text encoder with the original one to recover all unlearned concepts. Therefore, in open-source settings, fine-tuning the core denoising model (i.e., the U-Net) makes more sense and is a more robust approach. This can be seen in Table 4, where HiRM-R does not perform as well as ESD, CA on Flux architectures (which use two text encoders rather than one, as in Stable Diffusion)
-	The proposed method appears to be a simple extension of RMU (a popular unlearning method for LLMs) to diffusion models, with the additional trick of fine-tuning only the early layers instead of all layers. This could be viewed as the result of a simple ablation study on layer selection rather than a fundamentally new approach.
-	While the paper emphasizes its efficiency (faster than training-based methods), this seems somewhat expected — since it only updates weights without requiring output generation (as in ESD or other training-based methods), it is naturally much faster.

**Questions:**

•  Does the method employ any losses to preserve unrelated concepts, similar to UCE or SHS? If yes, how are the to-be-retained concepts chosen?

•  How does the method perform against the Random Probe recovery attack proposed in [1], which adds noise to the text encoder to confuse generation and recover unlearned concepts?

•  How are the super-categories determined? Are they predefined manually, or can they be learned end-to-end as in [2]?

[1] Lu, Kevin, et al. "When Are Concepts Erased From Diffusion Models?." NeurIPS 2025

[2] Bui, Anh, et al. "Fantastic targets for concept erasure in diffusion models and where to find them." ICLR 2025

---

> ### Author Response · Authors · 2025-11-23
> **Official Response to Reviewer X6qh  (1/3)**
>
> > [W1] In open-source settings, fine-tuning the core denoising model (i.e., the U-Net) is more sensible and offers a more robust approach than fine-tuning the text encoder.
>
> We thank the reviewer for raising this point about the limitations of text-encoder-only unlearning in open-source model release scenarios. We agree that if a safety-guarded model is released where only the text encoder has been modified, a malicious user can in principle swap the sanitized encoder with an off-the-shelf text encoder (e.g., CLIP) and thereby recover the erased concepts. In contrast, for a model whose core denoiser (e.g., U-Net) has been fine-tuned for unlearning, there is no publicly available “original” backbone that exactly matches the released checkpoint, so performing an analogous rollback is much less realistic in practice. In this sense, denoiser–based unlearning is preferred in the particular setting the reviewer describes.
>
> Our goal with HiRM is therefore not to replace denoiser-based unlearning in this fully open-source model release setting, but to **address a complementary regime**: provider- or platform-controlled deployments where (i) the model weights are not meant to be modified by end users, and (ii) a single text encoder is shared across many heterogeneous generators (different SD variants, LoRA-tuned models, Flux-like architectures, etc.). In such environments, text-encoder-side unlearning is more effective because **a single HiRM patch can be amortized across multiple backbones** (U-Net fine-tuning typically has to be repeated per generator).
>
> Regarding Table 4 and Flux specifically, we agree with the reviewer that Flux-specific denoiser fine-tuning (e.g., ESD, CA) can achieve stronger robustness on that particular architecture. This is expected: these methods are trained directly on Flux, while our HiRM-R encoder is **transferred zero-shot from an SD-1.4 text encoder without any Flux-specific tuning**. Despite this mismatch, HiRM-R still yields roughly a 50% reduction in NSFW/attack success rates compared to the original Flux model, which illustrates the transferability of an encoder-side patch across models with different denoisers and even dual-encoder designs.
>
> Moreover, we found that HiRM is best viewed as a **complementary component rather than a direct competitor** to denoiser–based approaches. When we combine HiRM-R with denoiser-based unlearning methods (CA, ESD, EraseAnything), we **consistently observe synergistic improvements**: attack success rates on benchmarks decrease substantially, while COCO-1k CLIP scores remain essentially unchanged. The table below summarizes these results:
> | Method | Ring-16 ↓ | Ring-38 ↓ | Ring-77 ↓ | MMA ↓ | I2P ↓ | COCO-1k (CLIP) ↑ |
> | :--- | :--- | :--- | :--- | :--- | :--- | :--- |
> | **CA** | 11.58 | 11.58 | 6.32% | 3.29 | 1.83 | **0.302** |
> | **CA + HiRM-R** | **3.16** | **2.11%** | **2.11** | 4.40 | **1.11** | **0.302** |
> | | | | | | | |
> | **ESD** | 41.05 | 33.68 | 32.63 | 7.10 | 2.93 | **0.307** |
> | **ESD + HiRM-R** | **12.63** | **7.37** | **4.21** | **3.30** | **2.51** | 0.306 |
> | | | | | | | |
> | **EraseAnything** | 29.47 | 24.21 | 26.32 | 6.60 | 2.64 | **0.305** |
> | **EA + HiRM-R** | **3.16** | **1.05** | **3.16** | **2.50** | **1.57** | **0.305** |
>
> These results indicate that an text-encoder-side HiRM patch can **strengthen existing denoiser–based defenses**, often yielding the **best overall erasure–utility trade-offs** among the tested configurations on these safety benchmarks, while preserving general image quality. We have included this discussion and the above table in Section 5.3 of the revised manuscript to make clear that HiRM’s main contribution is to provide a **lightweight, transferable encoder-editing mechanism that complements robust denoiser fine-tuning**, rather than to replace it in all deployment scenarios.

---

> ### Author Response · Authors · 2025-11-23
> **Official Response to Reviewer X6qh (2/3)**
>
> > [W2] The proposed method appears to be a simple extension of RMU to diffusion models with layer selection, which could be viewed as a simple ablation study rather than a fundamentally new approach.
>
> We appreciate the reviewer’s comment and agree that HiRM is conceptually related to Representation Misdirection (RMU) in that both steer internal representations toward alternative targets. However, our design goes beyond a direct “RMU but for diffusion” extension.
>
> First, HiRM explicitly **decouples the layer where we apply the misdirection loss from the layers we update**. RMU-style approaches typically apply the loss at a given depth and update that layer (and its nearby lower layers). In contrast, HiRM applies the loss on the **final** text-encoder block, where high-level concepts are formed, while restricting parameter updates to the **early blocks** only. This choice is informed by prior analyses showing that causal states for visual attributes in T2I models primarily reside in the early layers of CLIP-style encoders, whereas high-level semantics emerge in the last layers (Basu et al., 2023; Toker et al., 2024). Our existing ablations (Table 7) indicate that naive layer choices (e.g., updating only the last block or many upper blocks) lead to clear utility degradation, whereas the decoupled design yields a more favorable erasure–utility–robustness trade-off.
>
> Second, HiRM operates in a **retain-free regime**, which is a substantial difference with the original RMU formulation. In T2I, constructing a comprehensive retain set that covers the vast non-target concept space is unrealistic. Instead, we assess preservation using broad downstream metrics (COCO FID/CLIP, LPIPS, NSFW/adversarial ASR, MLLM-as-a-judge). Despite not using retain data, HiRM matches or exceeds the retention of methods that rely on curated retain sets (Tables 2–4), which we view as important for realistic deployment.
>
> Third, while RMU has mainly been instantiated with random targets, we investigate both **random** (HiRM-R) and **semantically guided** targets (HiRM-S). HiRM-S uses coarse guided concepts (e.g., super-categories such as *Painting*/*Animals* or NSFW safety directions), steering representations toward meaningful semantic regions rather than arbitrary noise. This improves the stability and plausibility of generations after erasure and supports better transfer when the same encoder patch is reused across different backbones (SD variants, Flux).
>
> In summary, HiRM builds on the idea of representation misdirection but tailors it to diffusion text encoders through (i) decoupled steering and update layers informed by causal analyses, (ii) a retain-free but empirically well-retained setting, and (iii) guided targets for stability and transfer. We therefore view HiRM as more than a simple layer-selection ablation of RMU.
>
> ---
>
> > W3] While the paper emphasizes its efficiency, this seems somewhat expected since it updates weights without requiring output generation, making it naturally much faster.
>
> We thank the reviewer for essentially pointing out one of the core practical advantages of our approach. We agree that, given HiRM’s design (no generative inner loop and only a small subset of encoder weights being updated), a speedup over training-based U-Net methods is expected. Our goal is not to claim “being fast” as a conceptual novelty, but to show that this expected efficiency comes **without giving up erasure and retention quality**.
>
> In practice, methods such as ESD, CA, and SalUn require repeated generation and per-backbone fine-tuning, which becomes prohibitive when maintaining many T2I models. Tables 2 and 3 show that HiRM attains comparable or stronger erasure–utility performance while avoiding any generative inner loop and updating only early encoder blocks, yielding significant speedup and substantially lower memory usage in our experiments.

---

> ### Author Response · Authors · 2025-11-23
> **Official Response to Reviewer X6qh (3/3)**
>
> > [Q1] Does the method employ any losses to preserve unrelated concepts, similar to UCE or SHS? If yes, how are the to-be-retained concepts chosen?
>
> We **do not use an explicit preservation loss** like UCE or SHS, and HiRM **does not rely on any external retain dataset**. This is a deliberate choice: in the T2I setting, constructing a comprehensive retain set that adequately covers the vast space of non-target concepts is impractical. Instead, HiRM controls over-unlearning through its **decoupled design** (steering at the last block while updating only early blocks) and is evaluated on broad downstream retention metrics (e.g., COCO FID/CLIP, LPIPS, MLLM-based judgments), which we found to be **sufficient to maintain unrelated concepts** in practice.
>
> ---
>
> > [Q2] How does the method perform against the Random Probe recovery attack proposed in [1], which adds noise to the text encoder to confuse generation and recover unlearned concepts?
>
> As suggested, we evaluated HiRM-S against Diffusion Completion (leveraging unfinished generations from the unerased base model) and Noise-based attacks (adding Gaussian noise to intermediate latents). The results demonstrate that HiRM-S achieves an **optimal trade-off**: it limits recovered Class Accuracy to 7.0% and 9.4% against Noise Probes  and Diffusion Completion, respectively, **significantly outperforming standard baselines** like ESD (21-37%) and UCE (42.7%). Crucially, this robustness does not come at the cost of utility; HiRM-S **maintains the highest accuracy on unrelated concepts** (81.1%), whereas other robust baselines such as GA and STEREO suffer from severe degradation (~52%).
>
> |  | Metric | UCE | ESD-x | ESD-u | GA | TaskVec | STEREO | RECE | HiRM-S |
> | :--- | :--- | :---: | :---: | :---: | :---: | :---: | :---: | :---: | :---: |
> | **Diffusion Completion** | CLIP (t=5) | 27.7 | 27.2 | 26.9 | 24.0 | **23.8** | 23.9 | 28.82 | 25.82 |
> | | Class Acc. (%) | 42.7 | 37.8 | 32.5 | **1.1** | 2.4 | 3.2 | 36.5 | 9.4 |
> | **Noise Based Probing** | CLIP | 27.8 | 28.0 | 27.7 | 26.1 | 26.5 | **24.6** | 27.0 | **24.6** |
> | | Class Acc. (%) | 2.67 | 21.9 | 30.7 | 27.7 | 11.0 | **1.1** | 13.0 | 7.0 |
> | **Unrelated Concept** | CLIP | 31.2 | 30.8 | 30.7 | 28.8 | 29.4 | 29.0 | 30.5 | **31.0** |
> | | Class Acc. (%) | 75.0 | 71.3 | 70.4 | 52.2 | 60.4 | 52.8 | 71.7 | **81.1** |
>
> We thank the reviewer for this valuable suggestion and have included these detailed results in Appendix B.5 of the revised manuscript.
>
> ---
>
> > [Q3] How are the super-categories determined? Are they predefined manually, or can they be learned end-to-end as in [2]?
>
> We appreciate the reviewer’s insightful question. In the current implementation, HiRM-S utilizes manually defined **super-categories** that align with the taxonomy of the underlying benchmark. For instance, specific style concepts (e.g., 'Van Gogh', 'Fauvism') are mapped to coarse groups like 'Painting', and object concepts (e.g., 'dogs') to broader categories like 'Animals'. Empirically, we found that such coarse groupings are sufficient to provide a stable semantic direction for misdirection, showing robust performance with low sensitivity to minor variations.
>
> We thank the reviewer for highlighting the end-to-end learning approach referenced in [2]. As HiRM is **agnostic to how guided concepts are obtained**, such methods are compatible with our framework and can be plugged in directly. While we consider the optimization of target discovery to be orthogonal to our main contribution (the decoupled encoder-editing strategy), we believe applying automated methodologies to discover effective guided concepts is a valuable and important direction for future work.
>
> ---
>
> [1] Lu, Kevin, et al. "When Are Concepts Erased From Diffusion Models?." NeurIPS 2025
>
> [2] Bui, Anh, et al. "Fantastic targets for concept erasure in diffusion models and where to find them." ICLR 2025

---

> ### Author Response · Authors · 2025-11-28
> **Official Response to Reviewer X6qh**
>
> Dear Reviewer X6qh,
>
> Thank you for taking the time to review our manuscript. We have carefully addressed all the comments and revised the manuscript accordingly. If there are any unresolved concerns or additional clarifications required, we would be glad to provide further details.
> We greatly value your feedback and look forward to hearing your thoughts.
>
> Best regards,
>
> The Authors

---

### Meta-Review · Area_Chair_uVNs · 2026-01-06

**Summary:**

This paper proposes an encoder-based concept erasure method that is incremental but technically sound. While initial concerns centered on limited novelty, robustness of text-encoder-only unlearning, missing baselines, and unclear evaluation, the authors’ rebuttal substantially addresses these issues. In particular, they clarify the deployment regime, position HiRM as complementary to denoiser-based methods, add missing comparisons (e.g., SPEED), and provide new experiments on robustness, multi-concept settings, and hybrid encoder–U-Net configurations. Although the gains over strong baselines are sometimes modest and the contribution is primarily empirical, the revised results demonstrate strong evidence with practical efficiency and transferability benefits. Overall, the paper marginally exceeds the acceptance threshold, making it a solid incremental contribution to the community.

**Reviewer Concerns:**

1. Limited Novelty / Incremental Contribution (R1~R4)
2. Marginal Performance Improvements and Weak Empirical Advantage (R2-R4)
3. Robustness and Security of Text-Encoder-Only Unlearning (R1 & R4)
4. Insufficient Coverage of Related Work and Baselines (R1, R2, & R4)
5. Evaluation Gaps and Incomplete Stress Testing (R2, R3, R4)
6. Lack of Theoretical or Interpretability Grounding (R1, R3, R4)

See further elaboration in the next part (Reviewer Scores).

**Reviewer Scores:**

R3 chose to maintain his positive rating of 6, and R4 raised his rating to 6 before the discussion stage was terminated.

For R1, the concerns of the robustness of text-encoder-only, table addition/flux discussion, and random probe attack are substantially addressed by the rebuttal responses. R1 might still be skeptical about the novelty (i.e., HiRM is an adaptation rather than a fundamentally new paradigm), and the efficiency claim is expected (as noted in the review). I feel R1 might be inclined to lean towards being positive about this work.

The AC also believes the rebuttal substantially addresses the core concerns of R2 (e.g., SPEED baseline and efficiency comparison, multi-concept erasure, Diff-Q and metric validity). Thus, R2 would likely upgrade to 6 or at least no longer oppose acceptance

---

### Decision · Program_Chairs · 2026-01-26

Accept (Poster)